# Innovative Biomedical and Technological Strategies for the Control of Bacterial Growth and Infections

**DOI:** 10.3390/biomedicines12010176

**Published:** 2024-01-13

**Authors:** Lídia Leonize Rodrigues Matias, Karla Suzanne Florentino da Silva Chaves Damasceno, Annemberg Salvino Pereira, Thaís Souza Passos, Ana Heloneida de Araujo Morais

**Affiliations:** 1Biochemistry and Molecular Biology Postgraduate Program, Biosciences Center, Federal University of Rio Grande do Norte, Natal 59078-970, RN, Brazil; lidialeonize@gmail.com; 2Nutrition Postgraduate Program, Center for Health Sciences, Federal University of Rio Grande do Norte, Natal 59078-970, RN, Brazil; karla.suzanne.damasceno@ufrn.br (K.S.F.d.S.C.D.); thais.passos@ufrn.br (T.S.P.); 3Nutrition Course, Center for Health Sciences, Federal University of Rio Grande do Norte, Natal 59078-970, RN, Brazil; annembergsalvino15@gmail.com

**Keywords:** antimicrobial agents, antibiotics, nanotechnology, genomics, bioinformatics

## Abstract

Antibiotics comprise one of the most successful groups of pharmaceutical products. Still, they have been associated with developing bacterial resistance, which has become one of the most severe problems threatening human health today. This context has prompted the development of new antibiotics or co-treatments using innovative tools to reverse the resistance context, combat infections, and offer promising antibacterial therapy. For the development of new alternatives, strategies, and/or antibiotics for controlling bacterial growth, it is necessary to know the target bacteria, their classification, morphological characteristics, the antibiotics currently used for therapies, and their respective mechanisms of action. In this regard, genomics, through the sequencing of bacterial genomes, has generated information on diverse genetic resources, aiding in the discovery of new molecules or antibiotic compounds. Nanotechnology has been applied to propose new antimicrobials, revitalize existing drug options, and use strategic encapsulating agents with their biochemical characteristics, making them more effective against various bacteria. Advanced knowledge in bacterial sequencing contributes to the construction of databases, resulting in advances in bioinformatics and the development of new antimicrobials. Moreover, it enables in silico antimicrobial susceptibility testing without the need to cultivate the pathogen, reducing costs and time. This review presents new antibiotics and biomedical and technological innovations studied in recent years to develop or improve natural or synthetic antimicrobial agents to reduce bacterial growth, promote well-being, and benefit users.

## 1. Introduction

Antibiotics are considered one of the great therapeutic advances of modern medicine. They contribute significantly to increasing life expectancy by treating various types of bacterial infections acquired by the world’s population. When used in conjunction with healthcare, they perform a critical role in the success of some medical practices [1,2].

The success of an antibacterial therapeutic agent, such as an antibiotic, is limited by the potential development of resistance in the target bacteria, compromising its effectiveness and resulting in the need for the development of new antibiotics or co-treatments [3,4].

Resistance to an antibiotic refers to an increase in tolerance to the therapeutic set or regimen to which the pathogen was susceptible before the emergence of resistance [3]. Thus, antibiotic-resistant bacteria can reduce or even fail the effect of antibiotics against infections caused by these microorganisms. Among bacteria, numerous Gram-positive and Gram-negative species have developed resistance to multiple classes of antibiotics [2].

In this context, antibiotic resistance has become one of the most severe and costly problems threatening human health in the 21st century [4,5]. In 2019, it was estimated that around 4.95 million deaths were associated with bacterial resistance, with approximately 1.27 million deaths worldwide attributed to bacterial resistance, with Europe and Africa having the lowest and highest mortality rates, respectively [6]. Among these deaths, the primary pathogens were *Escherichia coli*, *Staphylococcus aureus*, *Klebsiella pneumoniae*, *Streptococcus pneumoniae*, *Acinetobacter baumannii*, and *Pseudomonas aeruginosa*, responsible for 3.57 million associated deaths of antimicrobial resistance and 929,000 deaths attributable to antimicrobial resistance [6].

In the face of this scenario, the treatment of severe infections due to bacterial resistance is a challenge for healthcare professionals, considering that resistance may remain unrecognized until the identification of the causative agent and/or antimicrobial susceptibility testing [7]. This implies a non-negligible risk of delaying the initiation of active antibacterial therapy, with potentially unfavorable consequences in terms of survival and other outcomes [7].

The action of antibiotics occurs through different mechanisms: inhibition of synthesis of the microbial cell wall, inhibition of DNA synthesis or function, inhibition of synthesis of bacterial proteins, and lesion or destruction of membranes [8]. Therefore, a thorough understanding of these mechanisms is necessary to enable the development of strategies and innovations that can solve and overcome bacterial resistance to antibiotics.

New biomedical approaches to treatments have been developed to combat infections and bacterial resistance and advance the field of antibacterial therapies [9]. The molecular and structural understanding of bacterial resistance to antibiotics within the scope of bioinformatics is one of the areas explored for the discovery of antibacterial agents, as well as nanotechnology, promoting the development of new nanomaterials with antimicrobial and genomic potential to understand and provide information and resources in genetics for the control of bacterial growth and infections [10,11,12].

Therefore, this narrative review presents an updated and organized way of developing effective strategies for developing innovative tools in nanotechnology, genomics, and bioinformatics that have been evaluated to control bacterial growth.

## 2. Bacterial Resistant Mechanisms and Antibiotics

Pathogenic bacteria are studied in the context of infection models, observing that morphological diversity can influence the degree of colonization and pathogenicity. It is speculated that the cell shape may be a virulence factor or that the environment imposes a selective pressure leading to morphological diversity [13,14].

The bacteria can modify their morphology during their life cycle or in response to environmental conditions. Some bacteria enter a process of cellular differentiation called sporulation, culminating in the formation of a highly resistant spore that can survive for long periods under adverse environmental conditions such as high temperatures, radiation, and chemical aggression (Figure 1A) [14,15].

Gram-positive bacteria can cause severe and even fatal infections. There is worldwide concern regarding the resistance of Gram-positive pathogens, such as *Staphylococcus aureus* and *Staphylococcus epidermidis*, to methicillin due to prolonged and inadequate use of antibiotics [16].

Gram-negative bacteria are a major concern for researchers, as they are intrinsically resistant to many antibiotics considered effective against infections, such as glycopeptides, macrolides, aminocoumarins, rifamycins, and oxazolidinones. Although the intracellular targets of many antibiotics are present in Gram-negative bacteria, research has shown that the entry of these drugs into the cells is widely hindered by the outermost wall, requiring higher concentrations for growth inhibition than those achieved in clinical practice [17].

In addition, bacteria can survive environmental stress, and the evolution of antibiotic resistance is the most convincing evidence of this. The spore-forming bacteria may play an essential role in the evolution and spread of antibiotic resistance due to their ability to withstand antibiotic treatments, propensity for dispersal, and increased expression of genes encoding efflux pumps, which can also make bacteria less susceptible to antibiotics, depending on the bacterial strain [18]. Bacterial populations that have evolved independently and have been exposed to different antibiotics have shown variations in their genomic sequences whose functional consequences were identical or related [4].

Bacteria can acquire these variations in different ways: spontaneous genetic mutation, transmitted to offspring through Vertical Gene Transfer (VGT), and the acquisition of antibiotic resistance genes (ARGs) through Horizontal Gene Transfer (HGT), which occurs through the transmission of ARGs between different bacteria, even between other species [4] and in various types of HGT events that are well known as transformation, conjugation, and transduction [19] (Figure 1B). These variations reinforce the consequence of non-susceptibility to antibiotics used against bacteria [19]. Moreover, another alternative for antibiotic resistance is biofilm formation, which occurs with an assemblage of microbial cells through a polymeric matrix produced by themselves associated with a surface (Figure 1C) [20].

**Figure 1 biomedicines-12-00176-f001:**
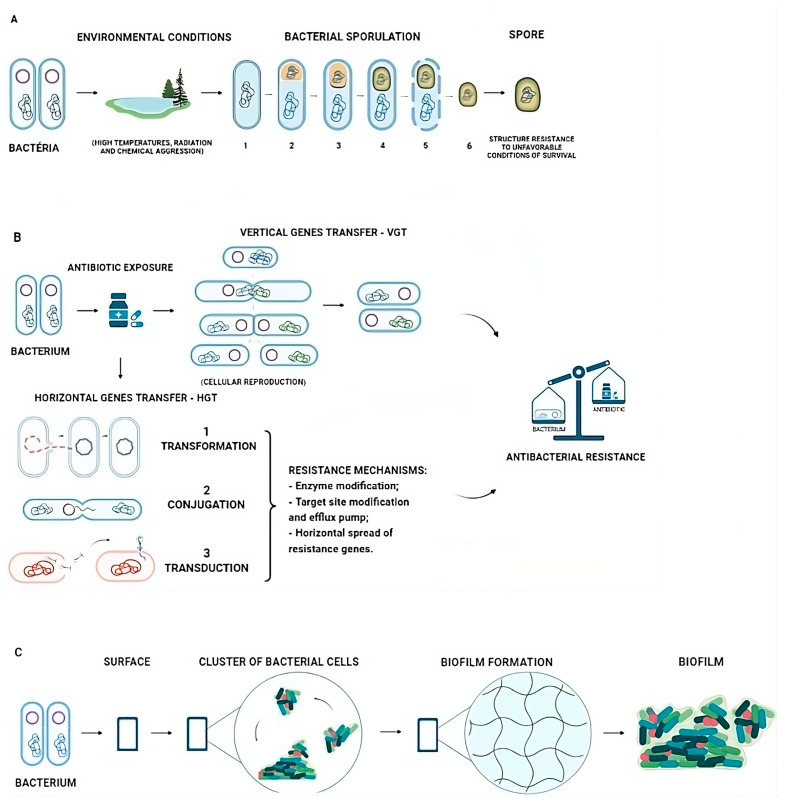
Variations acquired by bacteria responsible for developing antibiotic resistance. (**A**). A schematic representation of bacterial spore formation occurs when the bacteria are exposed to adverse and unfavorable extrinsic conditions: 1. Bacterial DNA. 2. Septum and forespore formation in the presence of the mother cell. 3. Engulfment (spore coat formation). 4. Complete spore (coat, outer forespore membrane, cortex, and inner forespore) 5. Mother cell lysis. 6. Endospore release. (**B**). VGT (Vertical Gene Transfer): the transfer of genetic material from parents to offspring (cell division). HGT (Horizontal Gene Transfer) [21]. 1. Transformation occurs with the acquisition of extracellular DNA incorporated into its genome and may develop the ability to express proteins associated with exogenous DNA or ARG (antibiotic resistance gene) [4]. 2. Conjugation describes the transfer of genetic material (usually plasmids) between two bacteria through cell-to-cell contact. 3. Transduction occurs when a bacteriophage particle transfers non-viral DNA from one bacterial host cell to another [22]. (**C**). Biofilm formation is represented by an inert or living surface, a self-produced matrix, or an assemblage of microbial cells associated with a surface [20].

Many species carrying resistance antibiotic enzymes belong to the phylum Actinobacteria [23]. One study showed that mobile resistance genes are mainly found in the phylum Proteobacteria, followed by Firmicutes, Bacteroidetes, and Actinobacteria. Researchers observed the transfer of the aph(3′)-III (aminoglycoside resistance), blaTEM-116 (beta-lactam resistance), tet(C) (tetracycline resistance), and catA (chloramphenicol resistance) genes among five or more different phyla. In these cases, phylogeny proved to be a determining barrier for HGT [24].

Regarding the mechanism of antibiotic resistance, bacteria from the Enterobacteriaceae family show broad resistance to β-lactam antibiotics, which are most commonly used throughout the world, due to the production of hydrolytic enzymes called β-lactamases. The resistance in Gram-negative bacilli to these antibiotics is due to acquiring plasmids containing genes that encode the synthesis of β-lactamases, especially in *Escherichia coli* and *Klebsiella pneumonia* [22,25].

In this context, it is necessary to comprehend antibiotics and their mechanisms of action to understand aspects related to resistance to combat bacterial infections. Therefore, antibiotics are natural or synthetic chemical substances primarily used to treat infections caused by pathogenic bacteria in humans and animals [26]. The activities of antibiotics on bacteria can be summarized into two modes of action, which involve diverse and distinct mechanisms: bactericidal, causing bacterial death, and bacteriostatic, inhibiting bacterial growth without causing death [1].

Antibiotics comprise the most successful group of pharmaceutical products, but the bacteria are subject to evolutionary evasion, promoting the development of resistance [26,27]. Additionally, the excessive use of antibiotics can lead to serious side effects and consequently increase the risk of antibiotic resistance dissemination. Since antibiotics do not distinguish between the microbiota and the harmful bacteria, despite the existence of narrow-spectrum antibiotics, antibiotic therapies can disrupt the intestinal microbiota, resulting in intestinal dysbiosis [28]. Regarding specificity, broad-spectrum antibiotics refer to those active against various microorganisms. As for narrow-spectrum antibiotics, they act on specific species of microorganisms [1].

The World Health Organization, in 2017, developed an antibiotic classification tool called AWaRe (Access, Watch, and Reserve) to support guidelines for antibiotic administration, emphasizing the importance of appropriate use of these drugs for each class of antibiotics (aminoglycosides; beta-lactams/beta-lactamase inhibitors; cephalosporins; lincosamide; metronidazole; nitrofurantoin; penicillin; trimethoprim/sulfamethoxazole; first-, third-, and fourth-generation cephalosporins; linezolid; monobactams; polymyxin; tigecycline; daptomycin; quinolones; macrolides; carbapenems; trimethoprim/sulfamethoxazole; and glycopeptides) [29]. As of 2021, the list of antibiotics covered 258, with 78 antibiotics added in four years [30].

The AWaRe aims to optimize the use of antibiotics and reduce antimicrobial resistance by classifying antibiotics into three groups. The Access Group has antibiotics with a narrow spectrum, indicated as the first alternative for most infections. The Watch group includes antibiotics with a broader spectrum, considered the second choice for treating infections. The Reserve group comprises antibiotics that should be used as a last therapeutic alternative in infections resistant to various drugs [30].

The most used classification of antibiotics is related to their mechanism of action [31], which occurs through modification of the structure and/or function of DNA of the cellular targets by decreasing the permeability of the outer membrane or through enzymatic inactivation or chemical modification as some examples of action from some antibiotics more used for infections treatments [1,17] (Table 1).

However, target bacteria may not respond to the action of antibiotics due to genetic mutations mentioned previously, whether through VGT or HGT, by acquiring ARG, and are capable of developing resistance mechanisms such as enzymatic modification, target site modification, efflux pumps, horizontal dissemination of resistance genes, mediated via transposons and plasmids, and expressing drug-insensitive variants of target enzymes [4,32].

Therefore, the discovery of molecules, enhancers, or adjuvant compounds that improve the activity of existing antibiotics, increase the permeability of the outermost membrane of Gram-negative bacteria, or avoid the modification of penicillin-binding proteins in Gram-positive bacteria and other possibilities that could reduce the bacteria’s infection is an innovative approach for studying new alternatives to combat bacterial resistance [17,34].

Nanotechnology, the combination of an antibiotic with another from a different class or some natural compound, using metagenomics and in silico studies to develop new effective strategies, appears as an innovative tool for promoting new approaches to combating bacterial resistance [17].

## 3. Innovations and Strategies in the Development of New Antibiotics

### 3.1. Use of Genomics and Resistant Genes for Antibiotics

Genomics is a field of science that studies genomes, evaluates the interaction between genes and the environment, and is considered a tool for researching and obtaining new natural products [36]. Given the increase in bacterial resistance to antibiotics and the reduction in substances previously capable of acting against pathogenic bacteria, a genomics area, such as metagenomics, proteomics, and transcriptomics, has been generating information about various genetic resources and assisting in discovering new molecules or antibiotic compounds [12,36,37]. Genomic technologies in routine biological research have brought the possibility of studying the metabolic pathways of microorganisms, which is important for studying the biosynthetic potential and appropriate cultivation, genetic sequencing, and conditions for bacteria that have not yet been known [38].

In 2016, Zipperer et al. [39] discovered a new antibiotic obtained from the bacterium *Staphylococcus lugdunensis*, which inhabits the human nasal cavity. Nasal strains of this bacterial species produce lugdunin. This new cyclic peptide antibiotic contains thiazolidine, a rare example of a bioactive compound synthesized non-ribosomally from bacteria associated with the human microbiota that inhibits colonization by *Staphylococcus aureus*.

In another study, the discovery of malacidins, a new class of antibiotics active against multidrug-resistant pathogens, was a positive and promising result of a soil microbiome screening method. In this discovery, Hover et al. [40] searched for calcium-dependent antibiotic molecules in the DNA of the soil microbiome, and genetic information from soil samples generated a database, validating a new class of antibiotics from natural resources. In this study, the genes from this genomic library were cloned and induced to produce biomolecules through fermentation; for example, malacidin effectively treats *Staphylococcus aureus* infections [40].

Another study with a relevant impact on the area was carried out, in which pneumococci were isolated from healthy individuals. Therefore, susceptibility to the antibiotics erythromycin and tetracycline was investigated, and the entire genome was sequenced. Therefore, the genetic context of resistance to one or both antibiotics was characterized. Tetracycline and macrolide resistance genes were detected in *Streptococcus. pneumoniae*. Thus, it is important to continue monitoring the antimicrobial resistance genes in vaccine and non-vaccine types in response to antimicrobial therapies and characterizing acquired resistance genes to continuously optimize antibiotic treatments and other studies to combat infections [41].

Harrison et al. [42] studied methicillin-resistant *Staphylococcus aureus* (MRSA), which is also resistant to almost all β-lactam antibiotics, limiting treatment options. Many isolates of this microorganism (MRSA) from different lineages were studied that are vulnerable to the action of penicillin used in combination with β-lactamase inhibitors such as clavulanic acid. This susceptibility is mediated through a combination of two different mutations in a promoter region that lowers mecA-encoded penicillin-binding protein 2a (PBP2a) expression and in isolates with either of the substitutions in PBP2a (E246G or M122I), increasing PBP2a’s affinity for penicillin in the presence of clavulanic acid.

Given this previous genomic study cited, an in vivo study in mice infected with *Staphylococcus aureus* showed that susceptibility to penicillin/β-lactamase, in the presence of clavulanic acid, can be exploited as an effective combinatorial therapeutic choice for MRSA infection. This fact suggests that susceptibility to penicillin/β-lactamase is an example of collateral sensitivity (resistance to one antibiotic increases sensitivity to another). Thus, it is suggested that antibiotics available on the market and currently disregarded may be effective in a significant proportion of MRSA infections [42].

According to Genilloud [43], natural antimicrobials are the origin of most classes of synthetic antibiotics currently in clinical use and continue to represent important molecules against infections resulting from natural evolution. Still, the rediscovery of known compounds has limited continued investment in natural product discovery programs.

One example is testing target-based approaches based on selective gene expression silencing, which involves sensitizing strains to a specific inhibitor targeting a pre-defined genetic product. This method has allowed the discovery of new classes of antibiotics, such as the FabF inhibitor platensimycin, kibdelomycin, and a new class of bacterial DNA gyrase inhibitors [43,44].

### 3.2. Use of Micro/Nanoparticles for Antibiotics

#### 3.2.1. Microparticles

A safe and efficient antibiotic administration system is essential to delivering therapeutic agents successfully [45,46]. Therefore, microparticles have emerged as low-cost alternatives with ease of synthesis, greater stability, and high drug transport capacity, leading them to be widely used as locally controlled drug delivery systems [46].

Some examples of microparticles containing antibiotics are from Rezić et al. [47], who developed silver (Ag) and zinc (Zn) particles, metals with potential antimicrobial activity, in different sizes (120 and 450 μm) and formulations with or without chitosan. For the microcapsule particles with 120 μm, the amounts of antimicrobial metals between the materials were similar. In contrast, microcapsules with 450 μm contained high enough amounts for efficient antimicrobial activity, enabling their application in different antimicrobial coatings and enhancing the effect of other encapsulated antibiotics [47].

Kost et al. [48] developed microparticles composed of homopolymer l-lactide and l-lactide/1,3-dioxolane, quercetin-loaded (co)polymers with sizes ranging from 60 to 80 µm, composed of semi-crystalline or amorphous degradable (co)polyesters. The microencapsulation of quercetin in a polymeric matrix enabled the prolonged release of the antimicrobial agent. In this study, the antibacterial properties of the biocompatible microparticles obtained were confirmed using the agar diffusion plate method. Their effectiveness was demonstrated against *Escherichia coli* and *Pseudomonas aeruginosa* bacterial strains, potentially having important applications in food preservation as a new antimicrobial system in the future.

Roque-Borda et al. [49] also evaluated the antibacterial activity of the antimicrobial peptide Ctx(Ile)-Ha obtained via solid-phase peptide synthesis (SPPS). The peptide exhibited potent antibacterial activity against *Salmonella enteritidis*, *Salmonella typhimurium*, *Acinetobacter baumannii*, and *Staphylococcus aureus*. Microencapsulation was highly efficient when the ionic gelation method was used. The hemolytic activity assay demonstrated that hemolysis decreased by up to 95% compared to the non-encapsulated active. Physicochemical stability was controlled with coated microcapsules. It was also shown that microencapsulation protected and adequately released Ctx(Ile 21)-Ha peptide in the intestine, making it a new and natural alternative for pathogen control.

#### 3.2.2. Nanoparticles

Nanoantibiotics are promising applications of nanotechnology. This technology encompasses the development of nanoparticles with antibiotics or pure antibiotics (without encapsulating agents) molecules synthesized artificially in a size range of ≤100 nm in at least one dimension [50]. This new area of antimicrobials revitalizes existing drug options, making them effective against various clinically significant bacteria with the help of antibiotic reengineering on a nanoscale [51].

Most conventional antibiotics require multiple doses in a systematic release. At the same time, nanoantibiotics bring the additional benefits of a controlled and specific release to the target, which can be administered in a single dose. Nanoparticles can present themselves as multifunctional “smart” antimicrobials that behave responsively to stimuli by interacting with the cell wall or surfaces of the bacterial membrane, leading to greater penetration through the membranes and drug delivery to target sites (Table 2) [51].

The antibacterial action of these nanomaterials (Figure 2) can be divided into aspects related to physical damage to the bacterial wall and membrane, chemical damage directed at bacterial cellular components, or synergistic damage to both membranes and intracellular components of bacteria [52]. In this context, nanomaterials such as metal oxide and polymeric nanoparticles have been used for antibiotic carrier applications [53].

The various nanomaterials with precious metals in their composition have stood out due to their optical, electronic, and catalytic properties [54]. The antibacterial effect of metal and metal oxide nanoparticles presents multi-target characteristics, as it alters the permeability of cell membranes as well as interferes with the functions of proteins containing sulfur and compounds containing phosphorus, such as DNA, making it difficult for bacteria to develop resistance to them [55].

Silver, gold, copper, iron, titanium dioxide, zinc oxide, etc., are the main metallic nano-antibacterial agents, and metal oxide nanoparticles such as silver oxide, iron oxide, copper oxide, zinc oxide, magnesium oxide, and bimetallic oxide are considered to have potential antibacterial activity, despite being excellent candidates as encapsulating agents for some antimicrobial or even antibacterial active compounds and being regarded as numerous investment alternatives [54].

A good example is the development of a nanoantibiotic capable of containing *Escherichia coli*’s resistance to the action of antibiotics. Silver nanoparticles are associated with the antibiotic ampicillin, comprising an optimized outer layer coated with silica [54]. Although the mechanism of action has not been completely clarified, it is believed that while the antibiotic acts through irreversible bonds to the serine amino acid in the active site of Penicillin-Binding Proteins (PBP), interrupting the synthesis of the cell wall and inhibiting enzymes such as transpeptidases and carboxypeptidases, silver nanoparticles act as a cytotoxic agent for them. On the other hand, when administered alone in the body, silver nanoparticles can result in undesirable effects, so new possibilities need to be studied [54].

Iron oxide nanoparticles have also been investigated as potential nanocarriers of antibiotics aiming to reduce infection. In a study, the potential of an iron oxide nanoconjugate containing the antibiotic teicoplanin was investigated against three Gram-positive bacteria (*Staphylococcus aureus*, *Enterococcus faecalis*, and *Bacillus subtilis*) and one Gram-negative bacteria (*Escherichia coli*). The results indicated that teicoplanin nanoparticles showed promising and prolonged antimicrobial activity against Gram-positive bacteria, unlike the isolated antibiotic [56].

Other applications using metal nanoparticles are related to gold nanoparticles containing daptomycin, which consist of a structure with intrinsic properties of both components that produce an enhanced bactericidal synergistic effect against *Staphylococcus aureus*, capable of breaking bacterial membranes by creating larger pores. These pores facilitate the entry of daptomycin nanoparticles into bacterial cells and contribute to more severe subcellular damage to bacteria, inducing high concentrations of reactive oxygen species (ROS) inside bacteria. These high ROS levels limit bacteria’s ability to develop drug resistance. This antibiotic–particle combination strategy provides a new perspective for synthesizing new antimicrobial agents [57].

One study with graphene oxide nanoparticles presents their morphology with sharp edges, known as nanoblades, which are responsible for bacterial death through physical contact with the membrane, generating numerous pores on the surface and contributing to the extravasation of intracellular substances and, consequently, cell death [52,58,59].

In this context, Chen, Wang, and Han [60] found that graphene oxide nanoparticles have superior bactericidal capacity against *Xanthomonas oryzae* pv. *oryzae* at low concentrations by rupturing the cell membrane, which was partially caused by the sharp edges of the nanoparticles. In addition, oxides can also act as peroxidases, catalyzing low concentrations of H_2_O_2_ into hydroxyl radicals (OH), demonstrating greater antibacterial activity, and avoiding biotoxicity due to the high H_2_O_2_ concentration [52].

Nanoparticles composed of polymers are one example, such as chitosan, a cationic, biodegradable, hydrophilic, and non-toxic polysaccharide that is a promising option as a nanocarrier for antibiotic drugs [61,62]. In one study, chitosan nanoparticles containing tetracycline, gentamicin, and ciprofloxacin were developed using the ionic gelation method. Some mixed cotton and cotton/polyester tissue samples were treated with nanoparticles of these antibiotics to evaluate their antibacterial activity. The results showed that tissues treated with chitosan nanoparticles and their respective nanocomposites effectively inhibited the growth of both Gram-positive and Gram-negative bacteria [63].

Another example of polymer nanoparticles is those composed of poly(lactic-co-glycolic acid) (PLGA), which have shown promise for drug-controlled and effective drug release due to their biodegradability and biocompatibility. The double-emulsion solvent evaporation technique synthesized PLGA nanoparticles conjugated with the antibiotics vancomycin and meropenem. The antibacterial action of the nanoparticles was effective and efficient when tested against *Staphylococcus aureus* and *Pseudomonas aeruginosa* [64].

Another technology within the field of nanotechnology that has gained great interest among researchers is the antimicrobial photodynamic inactivation of bacteria. This technique allows for controllable activation of antimicrobial effects by combining specific light excitation with the photodynamic properties of a photosensitizer, which generates ROS from molecular oxygen via electron or energy transfer, contributing to oxidative damage in nearby bacteria, suppressing their growth, and causing cell death [65,66].

As a more economical alternative, a system of brominated carbon nanoparticles has been developed as new photosensitizers for antimicrobial inactivation via photodynamics through combustion by-products collected with bromine incorporation in nanoparticles that allows for photosensitization effects via electron or energy transfer under UV-A irradiation. The efficacy of this new photosensitizer, still under study, is demonstrated by the growth inhibition reported for *Staphylococcus aureus*, *Listeria monocytogenes*, and *Escherichia coli* [66].

In addition, nanoantibiotics and nanoenzymes have broad-spectrum antimicrobial properties, with recent advances highlighting their antibacterial mechanisms and applications [67]. By combining the advantages of nanomaterials and natural enzymes, nanoenzymes are generally low-cost, have good stability, adjustable size, ease of preparation, a variety of functions, and superior catalytic activity. However, nanoenzymes have disadvantages, such as low efficiency in specific microenvironments, low selectivity and specificity, limited catalytic types, and potential nanotoxicity [68].

Nanomaterials are promising alternatives to overcome the problems of antibiotic therapies. However, studies point to the prospective use of nanotoxicity, limited knowledge of the interaction of nanomaterials with human cells, and the absence of characterization techniques that are not affected by the properties of nanoparticles [69,70,71]. Moreover, some microorganisms show resistance to metals and present some defense mechanisms such as extracellular sequestration, intracellular sequestration, active export (transport of heavy metal ions away from the intracellular environment is another process to defend against heavy metal stress), and enzymatic detoxification (controlled by microorganism defense genes) [72].

**Table 2 biomedicines-12-00176-t002:** Studies with nanotechnology applications related to antimicrobial activity.

Study	Encapsulating Agent	Antibiotic	Encapsulation Method	Particle Size	Bacteria(s) Target	Antibacterial Outcome
Oliveira et al., 2017 [54]	Silver nitrate (AgNO_3_)	Ampicillin	Colloidal dispersion	93 nm	*Escherichia coli*	The antibiotic interrupted the cell wall synthesis, inhibiting enzymes, and the silver nanoparticles acted as a cytotoxic agent for the bacteria.
Knoblauch et al., 2021 [66]	Brominated carbon (BrCND)	No antibiotic	Nanoincorporation (via carbon halogenation with bromine)	365 nm	*Staphylococcus aureus* and *Listeria monocytogenes* and *Escherichia coli*	The brominated carbon nanoparticles showed antimicrobial activity through the innovative method of photodynamic inactivation of bacteria with membrane rupture and release of reactive nitrogen species (synergical damage).
Armênia et al., 2018 [56]	Iron oxide (FeO)	Teicoplanin	Co-precipitation	10.5 nm	*Staphylococcus aureus, Enterococcus faecalis* and *Bacillus subtilis* and *Escherichia coli*	The isolated antibiotic presented antimicrobial activity in the short term, while the nanoparticles showed promising and prolonged antimicrobial activity because they caused a synergical effect with membrane and DNA damage.
Zheng et al., 2019 [57]	Gold (Au)	Daptomycin	Aggregation–Inducted Emission	190 nm	*Staphylococcus aureus*	The nanoparticles showed an enhanced bactericidal synergistic effect with the ability to disrupt bacterial membranes and produce ROS.
Chen et al., 2013 [60]	Graphene oxide (GO)	No antibiotic	Nanoprecipitation	300–600 nm	*Xanthomonas oryzae* pv. *oryzae*	Due to their sharp edges, the nanoparticles have shown superior bactericidal capacity at extremely low concentrations by rupturing the cell membrane.
El-Alfy et al., 2020 [63]	Chitosan	Tetracycline, Gentamicin, and Ciprofloxacin	Ionic gelation	3–4 nm	Gram-positive and Gram-negative bacteria	The three nanoantibiotics effectively acted on the inhibition of the growth of Gram-positive and Gram-negative bacteria due to chemical damage due to a charge imbalance.
Gaspar et al., 2018 [64]	Polylactic-co-glycolic acid (PLGA)	Vancomycin and Meropenem	Double emulsion–solvent evaporation	284.2 nm	*Staphylococcus aureus* and *Pseudomonas aeruginosa*	The nanoparticles of both antibiotics showed effective and efficient antibacterial action compared to the isolated actives, probably causing chemical damage.

### 3.3. Use of Computational Simulation for Antibiotics

The advances in bioinformatics have significantly contributed to genome sequencing, resulting in a large amount of data [73]. The innovative potential of using whole-genome sequencing of bacterial genomes for clinical diagnosis presents itself as a relevant strategy, considering the execution and improvement of this area [74]. The applications of this technology in molecular diagnosis can provide more precise epidemiological maps and information on the evolutionary history and genetic composition of specific microorganisms. In the clinical field, with the use of bioinformatics, it would be possible to conduct antimicrobial susceptibility tests without the need to culture the pathogen [74].

Additionally, the in silico computational approach makes it possible to select suitable enzymes and protein sources, perform proteolysis, predict possible biological activities, allergenicity, and toxicity, and determine mechanisms of action through molecular docking [75]. In silico analyses offer a valuable tool for searching for new drugs and existing information on an organism’s metabolites that are researched in available databases [76,77]. Importantly, in silico approaches offer the possibility of finding target compounds that inhibit disease processes, aiding in understanding the mechanism of action. Furthermore, these studies present advantages over in vitro studies, which often require time and significant resources and are vulnerable to failure [76,77].

Several commercially available antibiotics are non-ribosomal peptides (NRP), such as lugdunin, brevicidine, and laterocidine, and polyketides (PK), for instance, formicamycins, pyxidilicyline, talafun, and auroramycin. They are synthesized in large enzymatic complexes of non-ribosomal peptide synthase (NRPS) and polyketide synthase (PKS), modular enzymes that function like assembly lines [78,79]. Biosynthetic gene clusters (BGCs) encode these mega-enzymes in the bacterial genome.

Due to the antimicrobial properties of NRPS and PKS products, a lot of effort has been made to explore new NRP and PK to develop new approaches to combat the emerging resistance profile of pathogenic bacteria [79]. Genome mining is becoming essential for discovering new antibiotics by improving new sequencing technologies and bioinformatics software. This is due to the ability to quickly screen interesting bacterial genomes and metagenomes at a lower cost and with better efficiency, considering that these in silico approaches facilitate the identification of bacteria to be tested as a priority in vitro or in vivo [79,80].

In recent years, in silico tools have also become important for predicting toxicity and play a relevant role in regulation, decision making, and predictor selection in drug design, as in vitro/vivo methods are often limited by ethics, time, budget, and other resources [81]. In silico acute toxicity models are used for the research and development of products, product approval, registration, transportation, storage, and handling of chemicals. Thus, it becomes convenient to perform a preliminary assessment of the acute toxicity hazards of drugs, including antibiotics [82]. Moreover, it was discovered that there are still predictions that do not provide a clear understanding of biological pathways at the level of toxicity and, therefore, may not be suitable for determining toxicity [83].

A study with norfloxacin, ciprofloxacin, and sulfamethoxazole aimed to predict the toxicity in silico quantitative structure-activity relationship of their transformation/metabolism products and possible isomers. All compounds presented carcinogenic and non-biodegradable structures; only sulfamethoxazole presented a non-mutagenic structure. The results from the in silico models highlight the need to perform a risk assessment for transformation products and original antibiotics [84].

#### In Silico Studies and Antimicrobial Peptides (AMPs)

Antimicrobial peptides (*AMPs*) can be chemically synthesized or routinely produced by many bioinformatic strategies [85]. Thus, interaction databases are the most valuable tools for predicting the sites where peptides physically interact with bacterial membranes and optimizing the structure of motifs and charges by amino acid substitutions. Structurally modified *AMPs* are chemically synthesized or produced recombinantly, and their antimicrobial activities are evaluated based on optimized molecular interactions with the target pathogen [85].

*AMPs* with high hydrophobicity can aid in solid interaction with the lipid bilayer, a characteristic presented by microorganisms and a fundamental aspect used in computational studies [86,87]. On the other hand, peptides that contain cationic amino acids can facilitate the interaction of peptides with bacterial and negatively charged cellular membranes through electrostatic attraction [88].

Therefore, the antimicrobial activities of *AMPs* against bacteria and fungi are determined by a complex interaction between the average hydrophobicity, theoretical isoelectric point (pI), cationic, α-helix, and amphipathicity, which are crucial characteristics to define the biological activities of *AMPs* using computational models [86,89]. Regarding the three-dimensional structures, they influence the biological activities of *AMPs*. They are classified into four families: α-helix (β-sheet mixed structures and random coil) [90,91].

Thus, in silico design methods rely on a virtual bacterial membrane as a target for *AMPs*. This type of approach creates and/or modifies peptide sequences by increasing peptide cationic and hydrophobicity, mainly through the insertion of Lysine, Leucine, Alanine, or Glycine residues into the sequence, thereby enhancing the peptide-membrane interaction [92,93,94]. Therefore, it is possible to design new synthetic antimicrobial peptides by modifying the sequences (the peptide sequence’s complete or terminal ends) of these peptides found naturally in various organisms. It has been found that small changes in the amino acid composition can lead to changes in all the conformational and physicochemical properties of a peptide, including antimicrobial activity [95,96,97].

These studies can be promising in developing synthetic *AMPs*, reducing their size or the minimum inhibitory concentrations (MIC) for effective antimicrobial activity [94]. Moreover, modifications to the template peptide are usually carried out through truncation, amino acid substitution, hybridization, and/or cyclization [98,99]. In addition, most computational methods use a binary prediction and/or recognition configuration to assign an ‘*AMPs*’ or ‘non-*AMPs*’ label to a given peptide sequence [97,100].

Therefore, knowledge of structural diversity, as well as the increased presence of a specific amino acid at the complete or terminal ends of a peptide sequence, is essential for the design and production of synthetic analogs that are more active, selective, or chemically stable than natural *AMPs* and have great potential for clinical applications [94,101].

### 3.4. Other Strategies for Applying In Silico Studies

A concept called ‘hit compound’ is a strategy that can be expanded to meet the needs imposed by the threat of antibacterial resistance [102,103]. A hit compound is an active substance with reproducible activity, with a defined chemical structure (or set of structures), against one or more bacterial targets (polypharmacology), or a combination therapy in which the effects of several molecules are combined, which can be valuable [104,105].

Although the selectivity and cytotoxicity of hit compounds are recognized as important characteristics, their improvement should remain studied to optimize their effects against pathogens [105]. Depending on the target, hit combinations may act synergistically, preferably against different microorganisms, or additively [103,105]. An important approach to identifying successful new hit compounds is a high-throughput screening of virtual chemical libraries, and it is essential to select the correct set of compounds, such as a diverse set, a target-focused set, or a library of fragments of these compounds [103].

## 4. Natural Antimicrobials and New Strategies

Antimicrobial peptides (AMPs) can be used as an alternative in treating bacterial infections, and in addition, it is emphasized that they are part of the innate immune system [106]. AMPs are produced by bacteria, insects, amphibians, and mammals, and those chemically synthesized are potential candidates for designing and developing new antimicrobial agents. According to Sakthivel and Palani [107], antimicrobial peptides cannot have disulfide bridges. Still, they have a positive net charge and a hydrophobic character that facilitate the interaction with the membranes of microorganisms and can act by inhibiting their activity [108]. Some examples of AMPs are protease inhibitors, especially those from plant sources that have also shown antimicrobial capacity and promise for developing new therapeutics [109].

Among the alternatives, natural antibiotics stand out along with emerging technologies such as vaccines, antibody-antibiotic conjugates, probiotics, phage therapy, and diagnostics [1,110]. Some alternatives associated with antibiotics have been developed to promote greater safety against bacteria for consumers in various areas, such as food and clinical settings [111]. Especially noteworthy is the microbiological safety of food and the prevention and control measures for Waterborne and Foodborne Diseases (WFDs) [112,113].

### 4.1. Polymeric Films

Adding other food additives with antimicrobial properties is an option for active films and packaging [114]. Gul et al. [115] investigated the effects of films based on hazelnut flour protein nanoemulsion enriched with clove essential oil. The clove oil improved the antimicrobial activity of the edible films against *Listeria monocytogenes*, *Bacillus subtilis*, *Staphylococcus aureus*, *Pseudomonas aeruginosa*, and *Escherichia coli*.

Another example of a film with additives, according to Bourbon et al. [116], is carboxymethylcellulose-based films, which can act as a protective barrier on food surfaces, serving as carriers of bioactive compounds such as curcumin. This study also showed that protein-based nanohydrogels could be a good strategy for incorporating curcumin into edible films, highlighting their potential for use in food applications, mainly due to their antioxidant and antimicrobial activity, which could extend the shelf life of foods.

Films produced from biopolymers are alternatives to increasing the shelf life of food and reducing waste. Still, it may present aspects that make them sensitive to environmental conditions, in addition to their low mechanical resistance and potential characteristics for replacing non-degradable materials in the environment [117]. Other disadvantages are low mechanical and thermal resistance and high permeability to water vapor. This limits its use since effective control of water transfer is a desirable property for most foods, often requiring more than one polymer or plasticizing additive [118].

### 4.2. Alternative Medical Solutions

One example of a medical application is antibacterial peptide-based gels, which offer new clinical uses. They have excellent adhesion between implant surfaces and gums and present potential against infection development [119]. Another example is antibacterial peptides derived from crowberry used in urinary catheters, mainly to combat infections by *Staphylococcus aureus* and *Escherichia coli* [120].

Another alternative use of antibiotics is adhesive microcapsule dressings coated with polydopamine loaded with antibiotics. This dressing presents the capacity to adhere to the fibrous matrix to obtain a regulated release of ciprofloxacin to aid in the healing of dermatological eruptions and abrasions that present bacterial infections, in addition to the possibility of use in cases of bone and joint infections, unlike conventional dressings [121].

According to Gomes et al. [122], catheters are prone to bacterial adhesion and biofilm formation, contributing to infection and associated morbidity in hospitalized patients. Therefore, the effect of graphene nanoparticles that confer antimicrobial properties on silicone catheters was evaluated. The nanoparticles were successfully exposed to the surface of the silicone. These coatings induced greater bacterial death, demonstrating their potential use in the biomedical industry for silicone hospital materials to prevent deaths from infection.

Kabirian et al. [123] and Murungan and Rangasamy [124] highlighted that antimicrobial agents can also be incorporated with biomaterials to increase the antimicrobial activity of vascular grafts. However, it has been observed that these grafts also provide bactericidal activity against *Staphylococcus aureus* and *Escherichia coli*. Therefore, the presence of antimicrobial agents added to a biomaterial improves the quality of vascular grafts.

### 4.3. Probiotics

Antibiotics and probiotics used in combination have decreased the severity, duration, and occurrence of antibiotic-associated diarrhea without interruption of treatment due to collateral effects. This encourages people to follow their antibiotic prescriptions more closely to combat infection and not require extended periods of use, which prevents the spread of resistance related to inappropriate use of antibiotics. The extent to which probiotics directly prevent the transmission of antibiotic resistance is still being investigated; however, maintaining a healthy microbiome while taking antibiotics may have a chance to decrease the spread of resistance [125]. This may occur because probiotic bacteria have many beneficial properties to antagonize pathogenic bacteria. These properties include improving intestinal barrier function and immunity, competitive exclusion through reduced adhesion to cells, and co-aggregation, which in turn assist in eradicating the pathogenic organisms at the mucosa site [125].

Many studies point to probiotics and their metabolites (hydrogen peroxide, lactic acid, short-chain fatty acids, and bacteriocin) effectively reducing the risk of bacterial contamination [126,127]. Considering the potential of probiotics, combining probiotics with other antibacterial compounds can synergize against bacteria [128,129]. This combination may be a possible solution to control further biofilms’ growth, virulence, and formation [130]. Yang and Yang [130] conducted a microbiological study with *Clostridium difficile* subjected to a combination of antibiotics, including metronidazole, vancomycin, clindamycin, ceftazidime, or ampicillin, with *Bifidobacterium breve*. The results showed that *Bifidobacterium breve* could enhance the synergistic effect of these antibiotics through a combination of probiotic and antibiotic metabolites, promoting a greater antibacterial effect against Gram-negative or Gram-positive pathogens. However, the idea remains that probiotics are safer and more effective when they intervene after using antibiotics to combat infection, acting as an adjuvant [131].

### 4.4. Bacteriophages

According to Ling et al. [132], bacteriophages exhibit a significant bactericidal effect and are used to inhibit or kill harmful bacteria. The therapeutic effects of bacteriophages are similar for antibiotic-sensitive and resistant bacteria. Still, their inherent in vivo toxicity is low because they are primarily composed of nucleic acids and proteins, which are essentially non-toxic. Additionally, bacteria infected with bacteriophages are unable to recover their viability. Bacteriophages can also be bacteriostatic, which is relevant to inhibiting bacterial growth without the use of antibiotics, but this can be analyzed in detail with studies about the phylogenetic tree based on the phage large terminase subunit sequences, indicating a narrow relation between the phage and the specifical bacteria and according to the biological characteristics of the phage [133]. Therefore, bacteriophage therapy offers advantages over chemical antibiotics, such as increasing their numbers while killing bacteria. This reproductive capacity plays a positive role in antibacterial treatment.

An example of using phages to combat infections related to forming biofilms *Pseudomonas aeruginosa* or *Staphylococcus aureus* consists of infecting biofilms, promoting the lysis of the strains and the entire biofilm. Phages must be able to propagate within hosts in the biofilm matrix and release their antibacterial progeny into the medium conducive to dissemination to be effective against biofilms [134].

### 4.5. Vaccines

An alternative prophylactic to reducing antibiotic resistance is to invest in existing or new vaccines and vaccination strategies against global pathogens, such as *Streptococcus pneumoniae*, *Haemophilus influenza*, *Klebsiella pneumoniae*, and *Mycobacterium tuberculosis* [135]. Some broad areas of study for new vaccines include vaccinology, structural vaccinology, generalized modules for membrane antigens, bioconjugation, and adjuvants [135].

Some vaccines can also reduce antibiotic use by a proportion that exceeds the causal fraction of disease resulting from the target pathogen of the vaccine [136]. For example, an effective vaccine against group A Streptococcus would reduce the need for presumptive antibiotic treatment for pharyngitis, and vaccines against influenza would reduce antibiotic use days in adults by 28.1% [136,137]. Effective vaccines against key pathogens that cause a particular clinical syndrome can result in synergistic effects on antimicrobial use and thus reduce resistance [136].

Vaccines can be used with a much lower probability of resistance emerging than antibiotics [137]. They are used prophylactically and are effective before bacteria multiply after the initial infection (low pathogenic load) and before different tissues and organs are affected, reducing the possibility of mutations responsible for resistance appearing [138]. Increasing the coverage of existing vaccines and developing new vaccines targeting antibiotic-resistant organisms may play an important role, but new variants of microorganisms have rendered vaccines ineffective [139,140].

### 4.6. Clustered Regularly Interspaced Short Palindromic Repeats (CRISPR)

Another innovation in science that has been used is Clustered Regularly Interspaced Short Palindromic Repeats (CRISPR) systems, which have been studied to insert, delete, and mutate genes [141]. This tool has the potential for many applications in microbial engineering, including bacterial strain typing, culture immunization, autoimmunity or targeted cell death, and engineering or controlling metabolic pathways for improved biochemical synthesis [142].

However, implementing CRISPR-based antibacterials requires advances, including targeting specific pathogenic bacterial species in complex bacterial populations to deliver antibacterials to the pathogenic bacteria and, in some cases, providing these therapeutics to host cells infected with bacterial pathogens [141].

Delivery of CRISPR-Cas9 antibacterials presents a challenge as the active protein-RNA complex must pass through the bacterial membrane to be effective. Studies have skillfully used phage species specificity for CRISPR-Cas delivery, natural bacterial predators that efficiently inject DNA into bacteria [143]. Using phages specific to target species, bacterial chromosomal genes targeted by CRISPR-Cas9 can be encapsulated in phage capsids (protein coats), genetically encoding the machinery into a phagemid (DNA cloning vectors with bacteriophage and plasmid roles) [143].

An example of using this tool is presented in the study by Citorik, Mimee, and Lu [144], which transformed *Escherichia coli* with a plasmid-derived CRISPR-Cas9 targeting antibiotic resistance genes and a chromosomal copy of antibiotic resistance genes. Transformation of *Escherichia coli* with CRISPR-Cas9 guide RNAs targeting these genes decreased transformation efficiency in the presence of these plasmids. These promising in vitro results led to the use of phages to package vectors encoding CRISPR-Cas9 antibiotic-resistant genes. The addition of phage-encoded CRISPR-Cas9 resulted in the rapid death of specific bacteria.

Fuente-Núñez and Lu [145] studied CRISPR-Cas constructs designed to function as precision antimicrobials. They were shown to be capable of eliminating drug-resistant microbes, with CRISPR-Cas selectively targeting genes involved in antibiotic resistance. In contrast, it is necessary to determine the etiological factor causing an infection by culturing a clinical sample first and then identifying a pathogen using standard microbiology diagnostic procedures. This step is crucial to knowing the targeted bacterial species and the exact species [146].

These studies demonstrate that CRISPR-Cas9 antibacterials result in highly specific death of pathogenic bacteria, avoiding non-target species, an important criterion for developing new antibiotics. Furthermore, direct delivery of the CRISPR-Cas9 system into target cells using physical methods can be effective in cells cultured in vitro. Still, these physical methods are unsuitable for clinical application in vivo [147].

## 5. Conclusions

This narrative review presented innovative strategies and tools that successfully control bacterial growth. Nanotechnology, genomics, bioinformatics, and in silico studies have significantly optimized, developed, and enhanced antibiotics in the current market, considering the recovery and/or development of new antimicrobial agents.

In addition to the significant areas mentioned, we can also mention other alternative medical solutions such as the use of probiotics, bacteriophages, CRISPR, vaccines, AMPs, and polymeric films, all with strategies based on controlling the eradication of infections as well as possible sources of infection such as water, food, and hospital materials. Given this, numerous aspects can be addressed to reduce bacterial growth as a form of prevention, control, and even treatment, involving several technological and biomedical areas that are advanced today.

Notably, these strategies can be applied in different areas, whether in the pharmaceutical or food industries, to develop biomaterials or treat or prevent possible bacterial infections. Therefore, this review updates and presents the knowledge compiled to date related to developing or improving natural or synthetic antimicrobial agents through innovative tools and strategies clearly and objectively (Figure 3). Hopefully, this will inspire new studies and proposals in this specific area.

## Figures and Tables

**Figure 2 biomedicines-12-00176-f002:**
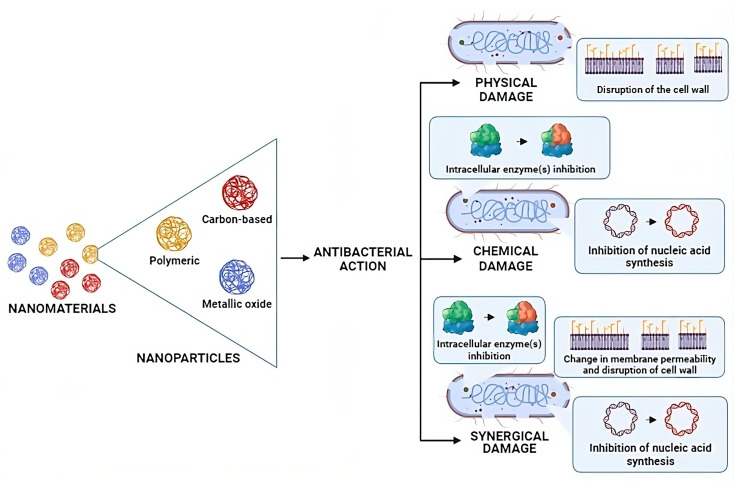
Antibacterial actions of nanomaterials in bacteria. One type of nanomaterial can act in a different antibacterial mechanism or use more than one. It depends on the target bacteria, the encapsulated agent’s combination, and the active compound(s) encapsulated. Physical damage: causes changes in the permeability and/or rupture of the bacterial membrane and wall. Chemical damage: it can inhibit important enzymes for bacterial metabolism, cause intracellular disorder due to electrostatic changes, and/or release reactive species of specific substances. Synergistic damage: physical and chemical damage occurring simultaneously.

**Figure 3 biomedicines-12-00176-f003:**
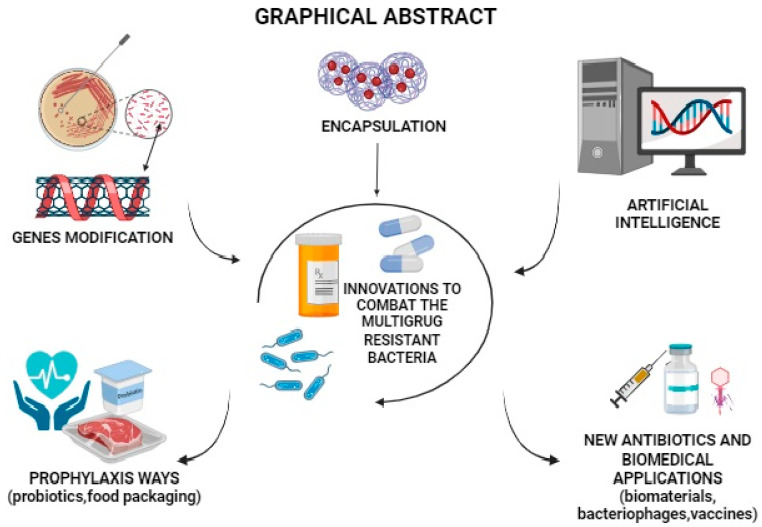
Nanotechnology, genomics, bioinformatics, and in silico studies contribute to the innovation of antibiotics.

**Table 1 biomedicines-12-00176-t001:** Action mechanisms of some antibiotics [32,33,34,35].

Mechanism of Action	Main Target Bacteria *	Main Antibiotics
**Cell Wall** They inhibit the synthesis of bacterial peptidoglycan cell walls. They act on enzymes called penicillin-binding proteins (PBPs) involved in cross-linking bacterial cell walls. The beta-lactam ring portion of these antibiotics irreversibly binds to PBPs, inhibiting the cross-linking of peptidoglycan and triggering bacterial death via autolysis.	*Staphylococcus aureus, Streptococcus pneumoniae, Listeria monocytogenes*, *Pseudomonas aeruginosa*, and Acinetobacter.	Penicillin, Carbapenem, and Cephalosporin
	*Staphylococcus epidermidis*, *Staphylococcus haemolyticus* and *Staphylococcus aureus*	Vancomycin and Teicoplanin
**Cell Membrane** They bind to phospholipids in the cytoplasmic membrane, altering their barrier function.	*Pseudomonas aeruginosa* *Staphylococcus aureus*, and *Bacillus subtilis*	Colomycin, Colistin, and Daptomycin
**Inhibition of Protein Synthesis** Binding to a susceptible organism’s 30S or 50S ribosomal subunit interferes with the binding of aminoacyl-tRNA to the mRNA/ribosome complex, thus interrupting bacterial protein synthesis.	Aerobic Gram-negative bacteria	Neomycin, Streptomycin, Kanamycin, Tobramycin, and Amikacin
		Oxytetracycline and Chlortetracycline
	*Streptococcus pneumoniae*, *Streptococcus pyogenes, Haemophilus influenzae* and *Moraxella catarrhalis*	Macrolides, Lincosamides, and Streptogramin B
		Chloramphenicol
**Nucleic Acids** They directly inhibit bacterial DNA synthesis by inhibiting two enzymes: topoisomerase II (DNA gyrase—catalyzes the negative supercoiling of closed double-stranded circular DNA) and topoisomerase IV (the unwinding of DNA after chromosomal duplication).	Gram-negative bacteria	Ciprofloxacin, Gemifloxacin, Levofloxacin, Moxifloxacin, and Ofloxacin

* It depends on the strain susceptibility profile.

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
