# Peer review of "Innovative Biomedical and Technological Strategies for the Control of Bacterial Growth and Infections"

_biomedicines, 2024, doi:10.3390/biomedicines12010176_

Round 1
Reviewer 1 Report
Comments and Suggestions for Authors
Well done for addressing all comments found in the previous submission
Author Response
Federal University of Rio Grande do Norte
Natal, December 28, 2023
Dear Editor at Biomedicines,
Thank you for considering our manuscript “Innovative biomedical and technological strategies for the control of bacterial growth and infections” (biomedicines-2695547) for resubmission. We have carefully read the reviewers' comments, answered all questions in this letter, and made the requested changes in the revised manuscript (highlighted in red). We have carefully addressed the manuscript's grammar, usage, and overall readability. A professional in English has revised the manuscript, and we also used the Grammarly software (full version) to check for gross mistakes further. Hopefully, this is a better version for revision. We are fully available to answer any further questions and attend to other suggestions. We hope the revised manuscript is suitable for publication in Biomedicines. This review is being submitted considering the invitation of Mr. Franklin Zhou (Section Managing Editor) to the special issue "Nanobiomaterials with Antimicrobial and Anticancer Applications".
Best regards,
Ana Heloneida de Araújo Morais
Department of Nutrition
Federal University of Rio Grande do Norte
Phone: (55 84) 991061887
E-mail: aharaujomorais@gmail.com or ana.morais@ufrn.br
Reviewers' comments:
Reviewer #1:
Comments and Suggestions for Authors
Well done for addressing all comments found in the previous submission.
- Thank you for considering our manuscript.

Reviewer 2 Report
Comments and Suggestions for Authors
Following my comments, the current review article has been partially revised, but more revision is needed.
1. “For the development of new alternatives, new strategies, and/or antibiotics for the control of bacterial growth, it is necessary to know the target bacteria, their classification, morphological characteristics, as well as the antibiotics currently used for therapies and their respective mechanisms of action, the information presented in this review.”
Remove “the information presented in this review.” This part does not fit well with this sentence.
2. Revise this sentence “through the sequencing of bacteria” as “through the sequencing of the bacterial genome.”
3. “Advances in bioinformatics have also significantly contributed to bacterial genome sequencing, resulting in a wealth of data.”
Advances in bacterial sequencing have resulted in a wealth of data and bioinformatics and have also contributed to developing new antimicrobials.
4. “In light of the above, this review aims to present new antibiotics and biomedical and technological innovations studied in recent years to develop or improve natural or synthetic antimicrobial agents to reduce bacterial growth, promote well-being, and benefit users, whether as a treatment for infections or prophylaxis.”
This sentence has a grammatical error. “whether as a treatment for infections or prophylaxis.” It does not fit well at the end of the sentence.
5. Although the manuscript has been revised to remove the aim for antibiotic resistance, almost all introduction is still on this issue (
6. “The action of antibiotics occurs through different mechanisms of action,” Two actions in a sentence need to be revised. Perhaps it would be better to remove the last action in the sentence.
7. “…all of which are effective against organisms and are effective against infections.” This sentence is not clear.
8. Sections A, B, and C in Figure 1 are not described in the figure legend. Only A is given in the legend. Moreover, I could not figure out what object represented the surface. Is it a mobile phone?
9. ” Microencapsulation was performed by highly efficient ionotropic.” This sentence is not clear.
10. “….related to physical damage to the bacterial membrane,” Bacterial membrane or bacterial cell wall? Figure 2 shows a damaged cell wall.
11. I do not understand what “Inhibition of protein synthesis in the nucleic acid means (Figure 2).
12. In Figure 2, “Change in the permeability e distribution of the membrane,” One part of the sentence is missing.
13. “Antibiotics and probiotics used in combination have been shown to decrease the severity, duration, and occurrence of antibiotic-associated diarrhea. This encourages people to follow their antibiotic prescriptions more closely, which slows the spread of resistance.” How can you associate the second sentence with the first one regarding the spread of resistance? Please clarify.
14. This title “4. Natural antibiotics and new strategies” better be revised as “4. Natural antimicrobials and new strategies”
15. The first paragraph of the conclusion is an introduction that could be removed.
Comments on the Quality of English LanguageIt is difficult to understand what is meant in many sentences, though it may be grammatically correct.
Author Response
Federal University of Rio Grande do Norte
Natal, December 28, 2023
Dear Editor at Biomedicines,
Thank you for considering our manuscript “Innovative biomedical and technological strategies for the control of bacterial growth and infections” (biomedicines-2695547) for resubmission. We have carefully read the reviewers' comments, answered all questions in this letter, and made the requested changes in the revised manuscript (highlighted in red). We have carefully addressed the manuscript's grammar, usage, and overall readability. A professional in English has revised the manuscript, and we also used the Grammarly software (full version) to check for gross mistakes further. Hopefully, this is a better version for revision. We are fully available to answer any further questions and attend to other suggestions. We hope the revised manuscript is suitable for publication in Biomedicines. This review is being submitted considering the invitation of Mr. Franklin Zhou (Section Managing Editor) to the special issue "Nanobiomaterials with Antimicrobial and Anticancer Applications".
Best regards,
Ana Heloneida de Araújo Morais
Department of Nutrition
Federal University of Rio Grande do Norte
Phone: (55 84) 991061887
E-mail: aharaujomorais@gmail.com or ana.morais@ufrn.br
Reviewers' comments:
Reviewer #2:
Following my comments, the current review article has been partially revised, but more revision is needed.
- “For the development of new alternatives, new strategies, and/or antibiotics for the control of bacterial growth, it is necessary to know the target bacteria, their classification, morphological characteristics, as well as the antibiotics currently used for therapies and their respective mechanisms of action, the information presented in this review.” Remove “the information presented in this review.” This part does not fit well with this sentence.
- We thank you for your care. The suggestion was accepted. We made the proper changes in the text, highlighting them in red.
- Revise this sentence “through the sequencing of bacteria” as “through the sequencing of the bacterial genome.”
- We thank you for your care. The suggestion was accepted. We made the proper changes in the text, highlighting them in red.
- “Advances in bioinformatics have also significantly contributed to bacterial genome sequencing, resulting in a wealth of data.”
Advances in bacterial sequencing have resulted in a wealth of data and bioinformatics and have also contributed to developing new antimicrobials.
- We thank you for your care. The suggestion was accepted. We made the proper changes in the text, highlighting them in red.
- “In light of the above, this review aims to present new antibiotics and biomedical and technological innovations studied in recent years to develop or improve natural or synthetic antimicrobial agents to reduce bacterial growth, promote well-being, and benefit users, whether as a treatment for infections or prophylaxis.”
This sentence has a grammatical error. “whether as a treatment for infections or prophylaxis.” It does not fit well at the end of the sentence.
- We thank you for your care. The suggestion was accepted. We made the proper changes in the text, highlighting them in red.
- Although the manuscript has been revised to remove the aim for antibiotic resistance, almost all introduction is still on this issue
- We appreciate the comment and reorganized the text before this submission. We reviewed the articles carefully, and all the mentions about antibiotic resistance are cited with reference. It is important to consider that one of the main problems to reduce the bacteria infection is the bacteria resistance. We made the proper changes in the text, highlighting them in red.
- “The action of antibiotics occurs through different mechanisms of action,” Two actions in a sentence need to be revised. Perhaps it would be better to remove the last action in the sentence.
- We thank you for your care. The suggestion was accepted. We made the proper changes in the text, highlighting them in red.
- “…all of which are effective against organisms and are effective against infections.” This sentence is not clear.
- We appreciate the comments, reorganized content throughout the text and considered the points mentioned in your review, highlighting them in red.
- Sections A, B, and C in Figure 1 are not described in the figure legend. Only A is given in the legend. Moreover, I could not figure out what object represented the surface. Is it a mobile phone?
- We thank you for your care. Figure 1 was reviewed. All the sections have subtitles mentioned. The object represents a surface, and we believe the changes done in the figure provided greater clarity.
- ” Microencapsulation was performed by highly efficient ionotropic.” This sentence is not clear.
- We thank you for your care. We considered the points mentioned in your review, highlighting them in red.
- “….related to physical damage to the bacterial membrane,” Bacterial membrane or bacterial cell wall? Figure 2 shows a damaged cell wall.
- We thank you for your care. We considered the points in your review and made changes in Figure 2.
- I do not understand what “Inhibition of protein synthesis in the nucleic acid means (Figure 2).
- We thank you for your care. We considered the points in your review and made changes in Figure 2.
- In Figure 2, “Change in the permeability e distribution of the membrane,” One part of the sentence is missing.
- We thank you for your care. We considered the points in your review and made changes in Figure 2.
- “Antibiotics and probiotics used in combination have been shown to decrease the severity, duration, and occurrence of antibiotic-associated diarrhea. This encourages people to follow their antibiotic prescriptions more closely, which slows the spread of resistance.” How can you associate the second sentence with the first one regarding the spread of resistance? Please clarify.
- We thank you for your care. The suggestion was accepted. We made the proper changes in the text, highlighting them in red.
- This title “4. Natural antibiotics and new strategies” better be revised as “4. Natural antimicrobials and new strategies”
- We thank you for your care. The suggestion was accepted. We made the proper changes in the text, highlighting them in red.
- The first paragraph of the conclusion is an introduction that could be removed.
- We thank you for your care. The suggestion was accepted. We made the proper changes in the text, highlighting them in red.
Comments on the Quality of English Language
It is difficult to understand what is meant in many sentences, though it may be grammatically correct.
- Thank you for your care. We reviewed all the text, considering your suggestion. We made the proper changes in the text, highlighting them in red.

Reviewer 3 Report
Comments and Suggestions for Authors
This work is about Innovative biomedical and technological strategies for the control of bacterial growth and infections.
The topic is actual and important, however, in order to be published, in my opinion the manuscript must be profoundly revised:
Major problems:
1) Page 2, Line 62: The sentence “The action of antibiotics occurs through different mechanisms of action, including chemical or mutational modification of cellular targets, decreased outer membrane permeability, efflux systems that expel the drug from inside the cell, and enzymatic inactivation of the drug through its breakdown or chemical modification.” Is completely wrong. These are mechanisms of resistance to antibiotics not mechanisms of action.
2) Page 2, Line 87: Spore formation is not a mechanism of resistance to antibiotics. It is mechanism of survival under unfavourable conditions meaning that bacteria are not proliferating (which is a condition to have an infection).
3) Page 4, Figure 1: Figure 1c is not coment on the text, therefore either it is useless or it should be comment.
4) Page 5: The text is not coherant and cohesive. It was not possible for me to follow the idea.
5) Page 6, Table 2: This information is disorganized and overly general. First, the mechanism of action should be presented. Within this, there is a vast variety. For example, it's not just 'protein' but rather 'inhibition of protein synthesis'. Among these, there are those that act on various subunits of ribosomes. Within antibiotics that act on the cell wall, we have the β-lactams and others
Additionally, only some examples have been chosen, not all, and this should be explained. The authors must also be careful because the target bacteria are not always the ones indicated. There are various species and strains of Gram-positive bacteria that are not susceptible to penicillin. This problem extends to the entire table.
6) Page 7, Line 228-235: The authors must clarify the resistance mechanisms they are referring to. Are they β-lactamases or alteration of PBPs? They are completely different mechanisms and this is not clear in the manuscript.
7) Page 13, Line 425: Give examples.
8) Page 15: AMPS were already discussed. Please reorganize ideas.
9) Page 15, Line 517: The sentence “Among the alternatives, natural antibiotics stand out along with areas emerging technologies such as vaccines, antibody-antibiotic conjugates, probiotics, phage therapy, and diagnostics” is a bit confusing. These are alternatives to antibiotics? How can vaccines and diagnostics can be considered alternatives to antibiotics? Please clarify.
10) Page 16, Lin2 595: The sentence “Bacteriophages can also be bacteriostatic, which is relevant in not promoting antimicrobial resistance already presented by many bacteria, but this can be analyzed with details with studies about phylogenetic tree based on the phage large terminase subunit sequences, indicating a narrow relation between the…” must be reformulated. Why being bacteriostatic enhance the absence of resistance?
11) Page 17, Line 609: The sentence “An alternative to reducing antibiotic resistance is to invest in existing or new vaccines and vaccination strategies” must be reformulated. See comment 9.
12) In general text should be reformulated in order to have a guiding thread. There is a lot of important information but it needs to be organized.
Minor problems:
1) I do not understand the yellow highlighting.
2) Page 2 Line 70: Where it is written “News biomedical approaches” should be “New biomedical approaches”
3) Page 14, Line 493: Where it is “.3.2.” should be “3.2”
The work must be reorganized and scientific comments addressed in order to be publish.
Author Response
Federal University of Rio Grande do Norte
Natal, December 28, 2023
Dear Editor at Biomedicines,
Thank you for considering our manuscript “Innovative biomedical and technological strategies for the control of bacterial growth and infections” (biomedicines-2695547) for resubmission. We have carefully read the reviewers' comments, answered all questions in this letter, and made the requested changes in the revised manuscript (highlighted in red). We have carefully addressed the manuscript's grammar, usage, and overall readability. A professional in English has revised the manuscript, and we also used the Grammarly software (full version) to check for gross mistakes further. Hopefully, this is a better version for revision. We are fully available to answer any further questions and attend to other suggestions. We hope the revised manuscript is suitable for publication in Biomedicines. This review is being submitted considering the invitation of Mr. Franklin Zhou (Section Managing Editor) to the special issue "Nanobiomaterials with Antimicrobial and Anticancer Applications".
Best regards,
Ana Heloneida de Araújo Morais
Department of Nutrition
Federal University of Rio Grande do Norte
Phone: (55 84) 991061887
E-mail: aharaujomorais@gmail.com or ana.morais@ufrn.br
Reviewers' comments:
Reviewer #3:
This work is about Innovative biomedical and technological strategies for the control of bacterial growth and infections.
The topic is actual and important, however, in order to be published, in my opinion the manuscript must be profoundly revised:
Major problems:
1) Page 2, Line 62: The sentence “The action of antibiotics occurs through different mechanisms of action, including chemical or mutational modification of cellular targets, decreased outer membrane permeability, efflux systems that expel the drug from inside the cell, and enzymatic inactivation of the drug through its breakdown or chemical modification.” Is completely wrong. These are mechanisms of resistance to antibiotics not mechanisms of action.
- We thank you for your care. The suggestion was accepted. We made the proper changes in the text, highlighting them in red.
2) Page 2, Line 87: Spore formation is not a mechanism of resistance to antibiotics. It is mechanism of survival under unfavourable conditions meaning that bacteria are not proliferating (which is a condition to have an infection).
- We thank you for your care. The topic of antibiotic resistance mechanisms brings up aspects regarding bacteria in general, not just related to infections in an isolated way. Still, some conditions can influence the development of infections. Moreover, for further clarification, a plausible reference was added to the text that presents spore formation as a mechanism of resistance to antibiotics, highlighted in red in the text.
3) Page 4, Figure 1: Figure 1c is not comments on the text, therefore either it is useless or it should be comment.
- We thank you for your care. We considered the points mentioned in your review, highlighting them in red.
4) Page 5: The text is not coherant and cohesive. It was not possible for me to follow the idea.
- Thank you for your care. The article was reviewed carefully. We made the proper changes in
the text, highlighting them in red.
5) Page 6, Table 2: This information is disorganized and overly general. First, the mechanism of action should be presented. Within this, there is a vast variety. For example, it's not just 'protein' but rather 'inhibition of protein synthesis'. Among these, there are those that act on various subunits of ribosomes. Within antibiotics that act on the cell wall, we have the β-lactams and others
Additionally, only some examples have been chosen, not all, and this should be explained. The authors must also be careful because the target bacteria are not always the ones indicated. There are various species and strains of Gram-positive bacteria that are not susceptible to penicillin. This problem extends to the entire table.
- We appreciate the comments. The table was reviewed, and the changes was highlighted in red in the table and the text.
6) Page 7, Line 228-235: The authors must clarify the resistance mec Whanisms they are referring to. Are they β-lactamases or alteration of PBPs? They are completely different mechanisms and this is not clear in the manuscript.
- We thank you for your care. The suggestion was accepted. We made the proper changes in the text, highlighting them in red.
7) Page 13, Line 425: Give examples.
R: We appreciate the comments, reorganized content throughout the text and considered the
points mentioned in your review, highlighting them in red.
8) Page 15: AMPS were already discussed. Please reorganize ideas.
- We thank you for your care. AMPs were cited in the article in two different contexts: in silico studies, using sequencing and remodeling tools for the development of antimicrobials, and studies of peptides that can be found in nature, which, when purified by physicochemical methods, can also act as antimicrobial agents. We made the proper changes in the text, highlighting them in red.
9) Page 15, Line 517: The sentence “Among the alternatives, natural antibiotics stand out along with areas emerging technologies such as vaccines, antibody-antibiotic conjugates, probiotics, phage therapy, and diagnostics” is a bit confusing. These are alternatives to antibiotics? How can vaccines and diagnostics can be considered alternatives to antibiotics? Please clarify.
- We appreciate the comments, reorganized content throughout the text and considered the points in your review, highlighting them in red.
10) Page 16, Lin2 595: The sentence “Bacteriophages can also be bacteriostatic, which is relevant in not promoting antimicrobial resistance already presented by many bacteria, but this can be analyzed with details with studies about phylogenetic tree based on the phage large terminase subunit sequences, indicating a narrow relation between the…” must be reformulated. Why being bacteriostatic enhance the absence of resistance?
- We appreciate the comment and reorganized content throughout the text according to your review and comments, making the text more precise and explaining better the sentence. The modifications are highlighted in red.
11) Page 17, Line 609: The sentence “An alternative to reducing antibiotic resistance is to invest in existing or new vaccines and vaccination strategies” must be reformulated. See comment 9.
- We thank you for your care. The suggestion was accepted. We made the proper changes in the text, highlighting them in red.
12) In general text should be reformulated in order to have a guiding thread. There is a lot of important information but it needs to be organized.
- Thank you for your care. The article was reviewed carefully. We made the proper changes in
the text, highlighting them in red.
Minor problems:
1) I do not understand the yellow highlighting.
2) Page 2 Line 70: Where it is written “News biomedical approaches” should be “New biomedical approaches”
3) Page 14, Line 493: Where it is “.3.2.” should be “3.2”
The work must be reorganized and scientific comments addressed in order to be publish.
- We thank you for your care. The suggestions were accepted. We made the proper changes in
the text, highlighting them in red.

Round 2
Reviewer 2 Report
Comments and Suggestions for Authors
Dear Authors,
The manuscript has been revised successfully following my comments. Only one sentence ("Advances in bacterial sequencing have resulted in a wealth of data and bioinformatics and have also contributed to developing new antimicrobials") is not corrected in the manuscript.
Regards.
Author Response
Federal University of Rio Grande do Norte
Natal, January 5, 2023
Dear Editor at Biomedicines,
Thank you for considering our manuscript “Innovative biomedical and technological strategies for the control of bacterial growth and infections” (biomedicines-2695547) for resubmission. We have carefully read the reviewers' comments, answered all questions in this letter, and made the requested changes in the revised manuscript (highlighted in yellow). We have carefully addressed the manuscript's grammar, usage, and overall readability. A professional in English has revised the manuscript, and we also used the Grammarly software (full version) to check for gross mistakes further. Hopefully, this is a better version for revision. We are fully available to answer any further questions and attend to other suggestions. We hope the revised manuscript is suitable for publication in Biomedicines. This review is being submitted considering the invitation of Mr. Franklin Zhou (Section Managing Editor) to the special issue "Nanobiomaterials with Antimicrobial and Anticancer Applications".
Best regards,
Ana Heloneida de Araújo Morais
Department of Nutrition
Federal University of Rio Grande do Norte
Phone: (55 84) 991061887
E-mail: aharaujomorais@gmail.com or ana.morais@ufrn.br
Reviewers' comments:
Comments and Suggestions for Authors ( #Round 2)
Reviewer #2:
Dear Authors,
The manuscript has been revised successfully following my comments. Only one sentence ("Advances in bacterial sequencing have resulted in a wealth of data and bioinformatics and have also contributed to developing new antimicrobials") is not corrected in the manuscript.
Regards.
- Thank you for considering our manuscript. We made the proper changes in the text, highlighting them in yellow.

Reviewer 3 Report
Comments and Suggestions for Authors
Some previous comments (in red) need to be further addressed:
1) Page 2, Line 62: The sentence “The action of antibiotics occurs through different mechanisms of action, including chemical or mutational modification of cellular targets, decreased outer membrane permeability, efflux systems that expel the drug from inside the cell, and enzymatic inactivation of the drug through its breakdown or chemical modification.” Is completely wrong. These are mechanisms of resistance to antibiotics not mechanisms of action.
- We thank you for your care. The suggestion was accepted. We made the proper changes in the text, highlighting them in red.
In fact the sentence was changed, however is still not correct. The sentence now is:
“The action of antibiotics occurs through different mechanisms, including decreased outer membrane permeability and inhibition of nucleic acid replication enzyme. Enzymatic mechanisms are considered the most efficient and can spread more easily among bacteria”.
Decreased outer membrane permeability is a resistance mechanism not a action mechanism.
Inhibition of nucleic acid replication enzyme is a mechanism of action.
Enzimatic mechanisms of what? I suppose authors refer to enzymatic mechanisms of resistance to antibiotics.
Authors must think properly about what they want to refer, mechanisms of action or mechanisms of resistance to antibiotics. In fact in Table 1 mechanism of action are referred and they are correct.
5) Page 6, Table 2: This information is disorganized and overly general. First, the mechanism of action should be presented. Within this, there is a vast variety. For example, it's not just 'protein' but rather 'inhibition of protein synthesis'. Among these, there are those that act on various subunits of ribosomes. Within antibiotics that act on the cell wall, we have the β-lactams and others
Additionally, only some examples have been chosen, not all, and this should be explained. The authors must also be careful because the target bacteria are not always the ones indicated. There are various species and strains of Gram-positive bacteria that are not susceptible to penicillin. This problem extends to the entire table.
- We appreciate the comments. The table was reviewed, and the changes was highlighted in red in the table and the text.
This is now Table 1. I still think that as the title is “Table 1. Action mechanisms of some antibiotics”, the first column should be mechanism of action. However I let the editor decide on that.
It is not “DNA” as mechanism of action. It is “Nucleic Acids”. There are antibiotics that inhibit RNA synthesis.
On the examples of bacteria, authors should put a note saying “MAIN TARGET BACTERIA (depending on the strain susceptibility profile)” instead of “SOME EXAMPLES OF MAIN TARGET BAC-TERIA”.
On the previous comment “only some examples have been chosen” was referred to the examples of antibiotics. That is also why, in my opinion, as these are examples of antibiotics and not all the antibiotics within each mechanisms of action I think this shouldn’t be the first column.
10) Page 16, Lin2 595: The sentence “Bacteriophages can also be bacteriostatic, which is relevant in not promoting antimicrobial resistance already presented by many bacteria, but this can be analyzed with details with studies about phylogenetic tree based on the phage large terminase subunit sequences, indicating a narrow relation between the…” must be reformulated. Why being bacteriostatic enhance the absence of resistance?
- We appreciate the comment and reorganized content throughout the text according to your review and comments, making the text more precise and explaining better the sentence. The modifications are highlighted in red.
The sentence was changed to “Bacteriophages can also be bacteriostatic, which is relevant to reducing the antimicrobial resistance already presented by many bacteria, but this can be analyzed with details with studies about phylogenetic tree based on the phage large terminase subunit sequences, indicating a narrow relation between the phage and the specifical 600 bacteria and according to the biological characteristics of phage [133]”. However, it is still not correct, because it is not the fact that bacteriophages are bacteriostatic that is relevant to reduce antimicrobial resistance. Ref 133 does not mention the fact that it is the bacteriostatic characteristic that allows the reducing of antimicrobial resistance.
Author Response
Federal University of Rio Grande do Norte
Natal, January 5, 2023
Dear Editor at Biomedicines,
Thank you for considering our manuscript “Innovative biomedical and technological strategies for the control of bacterial growth and infections” (biomedicines-2695547) for resubmission. We have carefully read the reviewers' comments, answered all questions in this letter, and made the requested changes in the revised manuscript (highlighted in yellow). We have carefully addressed the manuscript's grammar, usage, and overall readability. A professional in English has revised the manuscript, and we also used the Grammarly software (full version) to check for gross mistakes further. Hopefully, this is a better version for revision. We are fully available to answer any further questions and attend to other suggestions. We hope the revised manuscript is suitable for publication in Biomedicines. This review is being submitted considering the invitation of Mr. Franklin Zhou (Section Managing Editor) to the special issue "Nanobiomaterials with Antimicrobial and Anticancer Applications".
Best regards,
Ana Heloneida de Araújo Morais
Department of Nutrition
Federal University of Rio Grande do Norte
Phone: (55 84) 991061887
E-mail: aharaujomorais@gmail.com or ana.morais@ufrn.br
Reviewers' comments:
Comments and Suggestions for Authors ( #Round 2)
Reviewer #3:
1) Page 2, Line 62: The sentence “The action of antibiotics occurs through different mechanisms of action, including chemical or mutational modification of cellular targets, decreased outer membrane permeability, efflux systems that expel the drug from inside the cell, and enzymatic inactivation of the drug through its breakdown or chemical modification.” Is completely wrong. These are mechanisms of resistance to antibiotics not mechanisms of action.
R.We thank you for your care. The suggestion was accepted. We made the proper changes in the text, highlighting them in yellow.
In fact the sentence was changed, however is still not correct. The sentence now is:
“The action of antibiotics occurs through different mechanisms, including decreased outer membrane permeability and inhibition of nucleic acid replication enzyme. Enzymatic mechanisms are considered the most efficient and can spread more easily among bacteria”.
Decreased outer membrane permeability is a resistance mechanism not a action mechanism.
Inhibition of nucleic acid replication enzyme is a mechanism of action.
Enzimatic mechanisms of what? I suppose authors refer to enzymatic mechanisms of resistance to antibiotics.
Authors must think properly about what they want to refer, mechanisms of action or mechanisms of resistance to antibiotics. In fact in Table 1 mechanism of action are referred and they are correct.
R.We thank you for your care. The suggestion was accepted. We made the proper changes in the text, highlighting them in yellow.
5) Page 6, Table 2: This information is disorganized and overly general. First, the mechanism of action should be presented. Within this, there is a vast variety. For example, it's not just 'protein' but rather 'inhibition of protein synthesis'. Among these, there are those that act on various subunits of ribosomes. Within antibiotics that act on the cell wall, we have the β-lactams and others
Additionally, only some examples have been chosen, not all, and this should be explained. The authors must also be careful because the target bacteria are not always the ones indicated. There are various species and strains of Gram-positive bacteria that are not susceptible to penicillin. This problem extends to the entire table.
R.We appreciate the comments. The table was reviewed, and the changes was highlighted in yellow in the table and the text.
This is now Table 1. I still think that as the title is “Table 1. Action mechanisms of some antibiotics”, the first column should be mechanism of action. However I let the editor decide on that.
It is not “DNA” as mechanism of action. It is “Nucleic Acids”. There are antibiotics that inhibit RNA synthesis.
On the examples of bacteria, authors should put a note saying “MAIN TARGET BACTERIA (depending on the strain susceptibility profile)” instead of “SOME EXAMPLES OF MAIN TARGET BAC-TERIA”.
On the previous comment “only some examples have been chosen” was referred to the examples of antibiotics. That is also why, in my opinion, as these are examples of antibiotics and not all the antibiotics within each mechanisms of action I think this shouldn’t be the first column.
R. We appreciate the comment and reorganized content throughout Table 1, according to your review and comments, making the text more precise. The modifications are highlighted in yellow.
10) Page 16, Lin2 595: The sentence “Bacteriophages can also be bacteriostatic, which is relevant in not promoting antimicrobial resistance already presented by many bacteria, but this can be analyzed with details with studies about phylogenetic tree based on the phage large terminase subunit sequences, indicating a narrow relation between the…” must be reformulated. Why being bacteriostatic enhance the absence of resistance?
R.We appreciate the comment and reorganized content throughout the text according to your review and comments, making the text more precise and explaining better the sentence. The modifications are highlighted in yellow.
The sentence was changed to “Bacteriophages can also be bacteriostatic, which is relevant to reducing the antimicrobial resistance already presented by many bacteria, but this can be analyzed with details with studies about phylogenetic tree based on the phage large terminase subunit sequences, indicating a narrow relation between the phage and the specifical 600 bacteria and according to the biological characteristics of phage [133]”. However, it is still not correct, because it is not the fact that bacteriophages are bacteriostatic that is relevant to reduce antimicrobial resistance. Ref 133 does not mention the fact that it is the bacteriostatic characteristic that allows the reducing of antimicrobial resistance.
R. Thank you for your care. This sentence was reviewed and changed according to the reference cited. The modifications are highlighted in yellow.

Round 3
Reviewer 3 Report
Comments and Suggestions for Authors
The manuscript is now publishable.